# Role and Potential of Different T Helper Cell Subsets in Adoptive Cell Therapy

**DOI:** 10.3390/cancers15061650

**Published:** 2023-03-08

**Authors:** David Andreu-Sanz, Sebastian Kobold

**Affiliations:** 1Division of Clinical Pharmacology, Department of Medicine IV, Klinikum der Universität München, LMU Lindwurmstrasse 2a, 80337 Munich, Germany; 2German Cancer Consortium (DKTK), Partner Site Munich, 81675 Munich, Germany; 3Einheit für Klinische Pharmakologie (EKLiP), Helmholtz Munich, Research Center for Environmental Health (HMGU), 85764 Neuherberg, Germany

**Keywords:** adoptive cell therapy, CD4^+^ T cell, immunotherapy, cytokines

## Abstract

**Simple Summary:**

While most cancer immunotherapies have focused on CD8^+^ T cells as the main drivers of antitumor immunity, increasing evidence indicates that CD4^+^ T cells also play a central role in the elimination of tumors. Naïve CD4 T cells can be polarized towards different helper subsets, which can drastically affect the antitumor response through interactions with other cell types and the tumor microenvironment. In this review, we provide an overview of the role of different T helper subsets in the immune system, their implications in cancer immunology, and their applications in adoptive T-cell therapy.

**Abstract:**

Historically, CD8^+^ T cells have been considered the most relevant effector cells involved in the immune response against tumors and have therefore been the focus of most cancer immunotherapy approaches. However, CD4^+^ T cells and their secreted factors also play a crucial role in the tumor microenvironment and can orchestrate both pro- and antitumoral immune responses. Depending on the cytokine milieu to which they are exposed, CD4^+^ T cells can differentiate into several phenotypically different subsets with very divergent effects on tumor progression. In this review, we provide an overview of the current knowledge about the role of the different T helper subsets in the immune system, with special emphasis on their implication in antitumoral immune responses. Furthermore, we also summarize therapeutic applications of each subset and its associated cytokines in the adoptive cell therapy of cancer.

## 1. Introduction

Adoptive cell therapy (ACT) consists of the extraction of the patient’s own T cells to modify and expand them ex vivo and subsequently reinfuse them into the patient. ACT can be classified into three different types: tumor-infiltrating lymphocytes (TILs), genetically engineered T-cell receptors (TCRs), and chimeric antigen receptor (CAR) T cells [1]. TIL therapy is based on the ex vivo expansion of the patient’s T cells isolated from the tumor microenvironment. On the other hand, TCR- and CAR-based therapies use peripheral blood T cells, which are genetically modified with tumor-specific TCRs or CARs. One determining difference between these two approaches is that TCRs require MHC expression, while CAR T cells can exert their effector function in an MHC-independent manner [2]. The respective advance in CAR and TCR technologies and clinical applications have been extensively summarized elsewhere [3,4].

Most of the efforts in the field of tumor immunology have traditionally focused on the study of CD8^+^ T cells since their cytotoxic capacity makes them crucial for fighting tumors [5]. However, in the last years, increasing evidence suggests that CD4^+^ T cells also play a very important role in tumor immunity. Although they were initially thought to mainly help B cells and CD8^+^ T cells through cytokine production, the role of T helper cells in the immune system is much more complex [6]. Apart from their helper functions, CD4^+^ T cells can directly kill tumor cells through the release of granzyme B and perforin or by secreting effector cytokines such as tumor necrosis factor (TNF)-α or interferon (IFN)-γ [7]. Additionally, they can also induce apoptosis through the TNF-related apoptosis-inducing ligand (TRAIL) or Fas/Fas ligand pathways [8,9]. However, in antitumor immunity, the effect of T helper cells is thought to be mostly indirect through cytokine and costimulatory signals. CD4^+^ T cells transiently express CD40L, which binds CD40 in antigen-presenting cells (APC) to upregulate the expression of MHC molecules and the secretion of cytokines [10]. On the other hand, T helper cells also provide costimulatory signals to CD8^+^ T cells, which promote their effector and memory functions and impair activation-induced cell death [7,11]. Interestingly, some studies also suggest that CD4^+^ T cells can sometimes act as APCs and present peptides to CD8^+^ T cells [6]. Overall, although the direct effect of T helper cells on the tumor cells is limited, they are essential players of the tumor microenvironment for an effective antitumoral immune response.

In response to the cytokine milieu in the tumor microenvironment, naïve CD4 T cells can differentiate into several T helper subsets with very divergent effects on antitumoral immune responses [12] (Figure 1). Notably, these T helper subsets are plastic and, following the appropriate environmental cues, can switch their phenotype from one subset to another [13]. In this review, we discuss the function of different T helper subsets in antitumoral immunity and how these different subsets and their associated cytokines can be applied in adoptive T-cell therapy approaches.

## 2. Th1 Cells

T helper 1 (Th1) cells play a key role in the defense against intracellular bacteria, fungi, and viruses. Their differentiation is promoted mainly by interleukin (IL) 12 [14,15], which, together with IL-18 [16,17], induces the production of IFN-γ [18]. Additionally, IL-27 has also been shown to synergize with IL-12 to trigger the release of the latter [19]. The signaling of these cytokines leads to the activation of signal transducer and activator of transcription (STAT) protein family members STAT1 and STAT4, which promote the transcription of T-bet. T-bet is a transcription factor specific to the Th1 lineage and enhances the Th1 phenotype through upregulation of IL-12 receptor beta 2 and IFN-γ expression [20], which, at the same time, promotes expression of T-bet in an autocrine manner [21].

### 2.1. Th1 Cells and Cancer

Th1 cells have been consistently associated with a favorable disease outcome in different types of cancer [22]. This is due, in part, to the antitumoral mechanisms induced by IFN-γ, including the inhibition of metastasis and angiogenesis, and the induction of cancer cell apoptosis. Furthermore, IFN-γ polarizes macrophages towards an M1 phenotype, induces T regulatory (Treg) cell fragility, and promotes tumor senescence and dormancy [23]. These features make Th1 cells a very potent T helper subset for elimination of tumors. Elevated amounts of Th1 cells in the tumor microenvironment have been associated with a favorable prognosis in a wide range of tumor entities, including non-small cell lung cancer [24], ovarian cancer [25], breast cancer [26], melanoma [24], glioblastoma [27], colorectal cancer [28], and laryngeal carcinoma [29]. On the other hand, in some contexts of chronic inflammation, IFN-γ has also been described to facilitate tumor metastasis and immune escape [23].

### 2.2. Th1 Cells and Adoptive T Cell Therapy

Considering the correlation between Th1 cells and good prognosis in most cancer types, it is not surprising that many studies have attempted to harness Th1 cells to boost the efficacy of adoptive T-cell therapy. In a mouse model of glioblastoma, CD4^+^ CAR T cells with a Th1 phenotype were shown to persist and maintain their effector potency for longer compared to CD8^+^ CAR T cells [30]. Similarly, TCR-engineered T cells targeting the human telomerase reverse transcriptase also displayed features of the Th1 subset and exerted a potent antitumoral response in a xenograft mouse model [31]. In TIL therapy, Th1 cells recognizing a particular mutation in the ERBB2IP protein achieved tumor regression in a patient with multi-metastatic cholangiocarcinoma [32].

Following the success of adoptively transferred T cells with a Th1 phenotype in different ACT models, several T-cell engineering approaches have attempted to therapeutically harness the features of Th1 cells. For example, upregulating T-bet in CAR T cells promoted their differentiation towards a Th1 phenotype and improved their functionality in a mouse model of lymphoma [33] (Figure 2a). Another strategy to induce a certain phenotype in adoptively transferred T cells is through the signaling of cytokines promoting the differentiation of this subset, and these, in the case of Th1 cells, are mainly IL-12 and IL-18.

#### 2.2.1. Interleukin 12 Engineering to Induce Th1 Differentiation

A preclinical study of B-cell lymphoma suggested that IL-12-secreting CD19 CAR T cells increased their IFN-γ production and could eradicate tumor cells without the need for depleting chemotherapy through the recruitment of host immune cells [34] (Figure 2b). Consistent with these findings, a recent study in a mouse model of lymphoma has also shown that conjugating IL-12 on the surface of CAR T cells promotes their polarization towards a Th1 phenotype, which led to an improved cytotoxic capacity and survival of the mice (Figure 2d). This strategy also activated an endogenous Th1 response against other antigens within the tumor [35]. Intratumoral delivery of IL-12 in combination with CAR T cells also showed increased infiltration of Th1-like CD4^+^ T cells into the tumor microenvironment [36] (Figure 2e). Furthermore, in a xenograft model for ovarian cancer, CAR T cells targeting MUC-16ecto and constitutively secreting IL-12 have shown enhanced antitumoral efficacy and increased secretion of IFN-γ, suggesting a Th1 polarization of those CAR T cells [37]. However, in this study, it is unclear whether this effect was mediated by CD4^+^ or CD8^+^ T cells. This approach was unsuccessful in a clinical study with one patient (NCT02587689) [38], but it is currently being evaluated in an ongoing phase I clinical trial (NCT02498912) [39].

Despite the promising results achieved with CAR T cells constitutively secreting IL-12, they also present important limitations related to their toxicity [40] and induction of T-cell exhaustion [41]. To overcome these issues, several studies have attempted to reproduce this IL-12 signaling in a more localized manner. For instance, CAR T cells expressing IL-12 under the transcriptional control of the nuclear factor of activated T cells (NFAT) promoter showed reduced toxicity in preclinical models [42,43], even though this approach was not able to solve the toxicity issues in further clinical studies (NCT01236573) [44]. Other strategies that have been employed in mouse models to restrict the effects of IL-12 to the tumor site include the use of membrane-anchored IL-12 [45,46], the integration of IL-12 into the CAR exodomain [47], or IL-12 nanostimulant-engineered CAR T cells [48] (Figure 2d,f). Nevertheless, in most of these studies, it is unclear whether CD4^+^ or CD8^+^ T cells are responsible for the antitumoral response.

#### 2.2.2. Interleukin 18 Engineering to Induce Th1 Differentiation

Besides IL-12, multiple strategies have also attempted to exploit the antitumor capacity of IL-18 in adoptive T-cell therapy. The constitutive expression of IL-18 by CAR T cells enhances their antitumor capacity by improving their proliferative potential and polarizing them towards a Th1 effector phenotype [49,50,51] (Figure 2b). Furthermore, to limit the potentially toxic effects of a systemic IL-18 release, CAR T cells have also been engineered to express IL-18 under the transcriptional control of the NFAT promoter. This controlled release of IL-18 was also sufficient to enhance the effector cell property of CAR T cells and recruit macrophages to the tumor site [50,52].

Another strategy to achieve more localized cytokine signaling is the use of switch receptors. In mouse studies, a chimeric cytokine receptor with the extracellular domain from the GM-CSF and the intracellular portion of the IL-18 receptor provided CAR T cells with improved antitumor activity and better persistence upon repeated antigen stimulation [53] (Figure 2c).

Despite the promising preclinical results, the administration of IL-18 to patients with metastatic melanoma failed to show any clinical benefit as a monotherapy (NCT00107718) [54]. Further efforts to unravel possible limitations of IL-18 administration identified a soluble IL-18 binding protein (IL-18BP) that acts as a decoy receptor [55]. The IL-18BP binds IL-18 with an affinity three orders of magnitude higher than its native receptor, therefore sequestering it and preventing the signaling [55]. Detry and colleagues engineered a decoy-resistant IL-18 which has been tested in animal models and could provide a potential solution for this issue [56]. Other limitations might arise from a more pleiotropic impact of IL-18 on other immune cells, which might obscure Th1-directed IL-18 functions.

## 3. Th2 Cells

T helper 2 (Th2) cells play their main role in type 2 immune responses, which involve the eradication of extracellular pathogens, bacteria, and allergens. Through secretion of Th2-associated cytokines such as IL-4, IL-5, IL-9, IL-13, and IL-25, they promote humoral immune responses and recruit and activate eosinophils and mast cells [57]. The in vivo differentiation of naïve CD4 T cells into Th2 cells is primarily mediated by IL-4, which is secreted by other Th2 cells, in addition to basophils, eosinophils, mast cells, and NKT cells [58]. IL-4 signals through STAT6 [59], which upregulates the Th2 key transcription factor GATA3 [60]. In combination with STAT5, GATA3 drives IL-4 expression, which acts in a positive feedback loop to amplify the signal [58]. Although this is the canonical pathway for GATA3 activation, alternative IL-4 independent pathways have also been described [61].

### 3.1. Th2 Cells and Cancer

Th2-mediated immunity has been traditionally considered to promote tumor growth by inducing angiogenesis and counteracting Th1-mediated responses. Nevertheless, recent studies have suggested that some components of the type 2 immune response could also play an important role in tumor elimination [62]. Therefore, the role of Th2 cells in cancer is pleiotropic and context dependent.

Multiple reports have suggested that Th2 cells and their associated cytokines inhibit antitumoral immune responses. The presence of Th2 cells has been correlated with a poor disease prognosis in pancreatic [63], gastric [64], ovarian [65], cervical cancer [66], and melanoma [67]. However, the exact mechanisms by which Th2 cells can promote tumor growth are still not fully understood. Th2-associated cytokines have been shown to play a key role in the immune escape of tumors [68]. Moreover, IL-10 can also inhibit antigen process and presentation by dendritic cells and activate Treg cells [62]. In addition, IL-4 can also regulate the phenotype of tumor-associated macrophages and indirectly enhance the metastatic capacity of cancer cells [69].

On the other hand, several clinical reports contrast with those findings and suggest that Th2 cells may play a relevant role in tumor elimination. In Hodgkin lymphoma [70], chronic lymphocytic leukemia [71], and breast cancer [72], Th2 cells or their associated cytokines were correlated with a favorable prognosis. The mechanisms driving this phenotype could be related to IL-4, which promotes the infiltration of macrophages and eosinophils into the tumor site [73]. In addition, IL-4 has also been shown to induce apoptosis of breast cancer cells in vitro [74,75].

### 3.2. Th2 Cells and Adoptive T Cell Therapy

Due to the pleiotropic properties of Th2 cells in cancer progression, the application of Th2 cells in adoptive T-cell therapy has been rather limited so far. In a mouse model of lymphoma, Th2 cells were reported to have similar in vivo antitumoral activity compared to Th1 cells. However, the mechanism of action was different between the two subsets. While Th1 cells induced cellular immunity and lymphocyte infiltration, Th2 cells promoted inflammatory responses and tumor necrosis [76].

In a preclinical study of melanoma, adoptively transferred antigen-specific Th2 cells outperformed Th1 cells in terms of metastasis clearance. This process was dependent on recruitment of eosinophils to the tumor site and the activation of the IL-4/STAT6 signaling pathway [77]. Similar results were found in a mouse model of myeloma, where Th2 cells eradicated tumors by inducing a type 2 immune response and recruiting M2 macrophages to the tumor. The mechanism by which M2 macrophages can inhibit tumor growth involves the secretion of arginase, an enzyme metabolizing arginine, which is necessary for protein synthesis of cancer cells [78]. Consistent with these findings, Th2 cells have also been shown to have antitumor properties in colon and pancreatic cancer by inducing an innate immune response from eosinophils and macrophages [79].

To our knowledge, besides in vitro polarization of T cells towards a Th2 phenotype, there are no studies using further T-cell engineering strategies to achieve a Th2 polarization. However, since IL-4 is the main mediator of Th2 differentiation and is also involved in their antitumor effect, several studies have attempted to mimic this signaling in the tumor microenvironment. Tumors genetically engineered to secrete IL-4 were more immunogenic and more efficiently cleared in mouse models of renal cancer [80], colorectal cancer [80,81], spontaneous adenocarcinoma [82], colon carcinoma [83,84], fibrosarcoma [85,86], and melanoma [85]. Other studies have achieved similar results by injecting IL-4 into the tumor-draining lymph node [87] or directly into the tumor [73]. Several clinical studies have attempted to use IL-4 in the treatment of advanced tumors, but neither of them has been able to show any clinical efficacy [88,89,90]. It is important to note that, unlike in preclinical studies, IL-4 was delivered systemically in these clinical trials. Therefore, local delivery into the tumor microenvironment may be required to induce the Th2-mediated antitumoral response.

Even though IL-4 plays the main role in Th2 differentiation, IL-13 also has very similar biological functions. Following the same rationale as with IL-4, tumor cells have been engineered to secrete IL-13. These modified cancer cells were less tumorigenic thanks to the recruitment of myeloid cells, mostly neutrophils and macrophages, to the tumor microenvironment [91,92]. On the other hand, contrasting evidence suggests that IL-13 can also promote tumor growth by inhibiting the activity of cytotoxic lymphocytes and inhibiting the release of IFN-γ [93].

## 4. Th9 Cells

Th9 cells are a subset of CD4^+^ T cells characterized by the secretion of IL-9, as well as IL-10 and IL-21 [94]. As Th2 cells, their differentiation is dependent on the signaling of IL-4 through STAT6, but it additionally requires the presence of TGF-β [95]. Th9 cells are involved in type 2 immune responses and, along with Th2 cells, play a role in the elimination of extracellular parasites and the development of allergies [96].

The transcriptional regulation of Th9 cells is highly complex and is mediated by transcription factors involved in other T helper cell subsets [97]. IL-4 signaling drives the phosphorylation of STAT6, which leads to upregulation of GATA3 expression and repression of the TGF-β-mediated Foxp3 induction [98]. NFAT1 induces IL-9 secretion [99] together with PU.1 [100], which also suppresses Th2 differentiation [101]. Finally, SMAD2, SMAD4, and interferon regulatory factor 4 (IRF4) are also essential for Th9 differentiation [102,103].

### 4.1. Th9 Cells and Cancer

The role of Th9 cells in tumor development has been shown to be highly dependent on the tumor entity and varies between hematological and solid malignancies. In hematological tumors, most studies point towards a pro-tumoral role of Th9 cells [104,105]. Some of the mechanisms that have been proposed through which IL-9 could promote tumor growth include promoting the survival of lymphoma cells by reducing their oxidative stress [106,107] and inducing immunosuppression from Treg cells [108,109]. The expression of IL-9 in patients has been associated with large cell anaplastic lymphoma and Hodgkin’s lymphoma [110]. Similarly, IL-9 mRNA levels were higher in tumor cells derived from nasal natural killer/T-cell lymphoma compared to healthy controls [111].

Studies in solid tumors, on the other hand, contrast with the results in hematological malignancies. Particularly in melanoma, extensive evidence shows that Th9 cells exert potent antitumoral responses by recruiting dendritic cells to the tumor and inducing the differentiation of tumor-specific CD8^+^ T cells [112,113,114,115,116,117,118]. Moreover, in breast cancer, Th9 cells have been shown to play a role in antitumoral immunity due to the release of IL-9 and IL-21 [119]. On the other hand, in hepatocellular carcinoma, Th9 cells have been suggested to promote tumor progression through upregulation of CCL20 [120]. In lung cancer, Th9 cells also induce tumor growth by enhancing the proliferative and migratory capacity of lung cancer cells [121].

### 4.2. Th9 Cells and Adoptive T-Cell Therapy

Despite the evidence showing that Th9 cells can exert potent antitumoral activities, their applications in adoptive T-cell therapy are still relatively scarce. In preclinical models of melanoma and lung adenocarcinoma, the adoptive transfer of antigen-specific Th9 cells has been shown to induce a potent antitumoral response [113,114,115,117]. Such antitumoral response is mediated by IL-9 secreted by Th9 cells and involves the recruitment and survival of dendritic cells to the tumor site [113,117]. Even though Th9 cells represent the main source of IL-9, it is important to note that other CD4 T-cell subsets such as Th2, Th17, and Treg can also produce IL-9 in smaller amounts than Th9. Moreover, some CD8^+^ cells, as well as mast cells and NKT cells, can also be a source of IL-9 [95].

Following the success of adoptive transfer of Th9 cells as a cancer treatment, some studies have tried to induce this phenotype in CAR T cells. Liu and colleagues showed that polarizing CAR T cells to secrete IL-9 improves their antitumoral capacity compared to conventional Th1-polarized CAR T cells [122] (Figure 3a). Those Th9-polarized CAR T cells exhibited an increased proliferative capacity, reduced levels of exhaustion, and a central memory phenotype. Interestingly, IL-9-secreting CAR T cells switch their cytokine secretion profile towards Th1 in vivo and produce IFN-γ. A recent report also showed that an orthogonal receptor with an extracellular domain of the IL-2 receptor and an intracellular domain of the IL-9 receptor increases the antitumor activity of T cells and polarizes them towards a stem cell memory and effector phenotype [123] (Figure 3b). In addition, in a mouse model of leukemia, in vitro polarized Th9 cells for allogenic bone-marrow transplantation achieved tumor eradication without inducing graft-versus-host disease [124].

## 5. Th17 Cells

Th17 cells are a subset of CD4^+^ T cells which secrete IL-17A, IL-17F, IL-21, and IL-22 [125]. Their main physiological function consists of providing host defense against extracellular pathogens at the mucosal surfaces. The key cytokines driving the differentiation of Th17 cells are TGF-β, IL-6, and IL-1β, in addition to IL-21 and IL-23, which are involved in the long-term maintenance of the Th17 phenotype [126]. The main transcription factor driving the differentiation of Th17 cells is the retinoic acid receptor-related orphan nuclear receptor gamma (RORγ) [127]. A unique feature of RORγ as a transcription factor, as compared to T-bet or GATA3, for example, is that its expression is not stabilized by positive feedback loops, making it extremely susceptible to changes in the environment [128]. At the cellular level, these fluctuations are translated into high phenotypic plasticity. Of note, there have been phenotypical transitions reported between Th17 cells and all other T helper subtypes [129].

### 5.1. Th17 Cells and Cancer

Considering the high phenotypic plasticity of Th17 cells, it is not surprising that the current evidence for the function of Th17 cells in cancer is highly inconsistent and dependent on the characteristics of the tumor. Since the differentiation pathways of Th17 and Treg cells are reciprocally regulated and are antagonistic from one to the other [130], many studies have attempted to correlate the ratio of Th17 and Treg cells with disease outcomes. Notably, an imbalance towards any of the two subsets could favor tumor progression, either due to an excessive inflammatory response or an excessive inhibition of the antitumoral immune response. Multiple studies have successfully correlated the imbalance between the Th17 and Treg populations with the cancer grade and survival of the patients [126,131,132,133,134], but the direction of such imbalance varies depending on the tumor types [135]. Th17 cells have been shown to facilitate tumor growth by inducing angiogenesis and promoting the survival of cancer cells. However, they can also play a role in tumor elimination by releasing IFN-γ and recruiting dendritic cells, CD8^+^ T cells, and natural killer cells to the tumor site [136]. The interaction between Th17 cells and cancer is therefore very complex and context dependent.

### 5.2. Th17 Cells and Adoptive T Cell Therapy

Several animal studies have suggested that adoptively transferred Th17 cells could be beneficial for tumor regression even to a greater extent than Th1 cells [137,138,139,140], although the exact mechanism is still unclear. Similar to Th9-polarized CAR T cells, when Th17 cells are injected in vivo, they acquire a memory phenotype and show enhanced persistence [138]. Furthermore, the antitumoral effect of Th17 cells in vivo has been shown to be dependent on IFN-γ, even though this cytokine is characteristic of the Th1 subset. This suggests that the plasticity of Th17 cells is essential for their implication in the antitumoral immune response since they polarize in vivo towards a Th1-like phenotype and secrete IFN-γ [137,138]. Despite the promising results of the use of Th17 as a potential tumor therapy in mouse models, this approach has not been used in clinical trials yet. This is, in part, due to the technical difficulties in efficiently generating enough Th17 cells without losing their antitumoral properties [126].

Given the interesting facets of Th17 cells in tumor immunity, some studies have attempted to exploit this phenotype to improve the functionality of CAR T cells. A chimeric cytokine receptor with the extracellular domain of the IL-4 receptor and the intracellular domain of the IL-21 receptor was able to polarize CAR T cells towards a Th17-like phenotype (Figure 4a). In the presence of IL-4, this switch receptor enhanced the killing capacity, as well as the in vivo persistence, of CAR T cells [141]. Furthermore, the costimulatory domains of the CAR have also been shown to change the polarization of T cells. CAR T cells containing the inducible co-stimulator (ICOS) intracellular domain polarized towards a Th17/Th1 phenotype and showed an increased in vivo persistence compared to CAR T cells with CD28 or 4-1BB costimulatory domains [142] (Figure 4b). A mutation in a residue of the CD28 costimulatory domain has also been shown to skew the polarization of CAR T cells towards Th17 (Figure 4b). This change in a single amino acid was able to reduce the differentiation and exhaustion of CAR T cells and improve their in vivo persistence and antitumoral activity [143].

Another strategy that has been employed to increase the infiltration of Th17 cells in the tumor site is through modifying the tumor microenvironment. Together with TGF-β, IL-6 is required to drive the polarization of Th17 cells. Gnerlich and colleagues engineered Pan02 cells, which already produce TGF-β physiologically, to secrete IL-6 (Figure 4c). These engineered tumor cells were able to reshape the tumor microenvironment and increase the number of Th17 cells, which resulted in reduced tumor growth and increased survival [144]. Nevertheless, the use of IL-6 in adoptive T-cell therapy approaches raises important safety concerns, since it is an important mediator of cytokine release syndrome (CRS) and may also promote tumor growth under certain circumstances. Therefore, upregulating the expression of IL-6 would be highly problematic in clinical applications.

## 6. Th22 Cells

Th22 cells are a more recently identified T helper subset characterized by secretion of IL-22, IL-13, and TNF-α to the exclusion of IL-17, IFN-γ, or IL-4 [145]. Their differentiation is mediated by the transcription factor aryl hydrocarbon receptor (AHR) [146]. The primary physiological role of Th22 cells is to protect epithelial barrier organs such as the skin and lungs and to modulate inflamed and injured tissue [147].

### Th22 Cells and Cancer

So far, Th22 cells and their main signature, cytokine IL-22, have been associated with tumor progression in patients with hepatocellular carcinoma [148], colon cancer [149], lung cancer [150,151], gastric cancer [152], and ovarian cancer [153], among others. This is probably due to the effect of IL-22, which has been shown to inhibit apoptosis and promote tumor cell proliferation, angiogenesis, migration, epithelial-to-mesenchymal transition, and metastasis [148,151,154,155,156,157,158]. However, it is important to consider that IL-22 is not exclusively produced by Th22 cells; it is also produced by Th1 and Th17 [159,160].

In a mouse model of neck squamous cell carcinoma, fourth-generation MUC1-targetting CAR T cells secreting IL-22 were shown to be more effective than the corresponding second-generation CAR T cells. Even though IL-22-secreting CAR T cells showed higher in vivo persistence, the main reason for the improved antitumoral activity was that IL-22 upregulates MUC1 expression in cancer cells, allowing a more effective recognition by CAR T cells [161]. Apart from this particular case, since IL-22 has mainly tumor-promoting properties, there are no other relevant applications of Th22 cells in adoptive cell therapy.

## 7. T Follicular Helper Cells

T follicular helper (Tfh) cells are a subtype of T helper cells whose main function is to support B-cell proliferation, somatic hypermutation, and class-switch recombination [162]. While the rest of T helper subsets described above travel to the site of infection or inflammation to exert their effector function, Tfh cells remain in lymph nodes and spleen to help in the process of B-cell development [163]. They are characterized by their expression of CXCR5, which is the receptor of CXCL13, and the transcription factor BCL-6 [164,165]. Differentiation of Tfh cells depends on IL-6 and IL-21, which activate STAT3 to upregulate the expression of BCL-6 [166].

### Tfh Cells and Cancer

In chronic lymphocytic leukemia, the presence of circulating Tfh-like cells has been reported as a predictor of poor prognosis, especially during advanced stages of disease [167,168]. Conversely, in non-lymphoid tumors, Tfh cells could be exerting an antitumoral function. In breast, lung, and colorectal cancer, infiltration of Tfh cells has been associated with a favorable prognosis and increased survival [169,170,171]. It has been hypothesized that the mechanism by which Tfh cells contribute to tumor elimination is through the development of ectopic lymphoid structures, which recruit other immune cell types involved in antitumoral immunity. Furthermore, through their implication in B-cell function, they could also be promoting antitumoral antibody responses [163]. In any case, there are still many open questions about the role of Tfh cells in cancer immunology, and further understanding of the mechanisms involved is required to harness Tfh cells as a cancer therapy. Adoptive transfer of Tfh cells has been shown to mediate antitumor immune responses in a CD8^+^ and IFNγ-dependent manner, with comparable efficacy to Th1 cells [172]. However, to our knowledge, there have not been any other relevant studies using Tfh-polarized T cells in adoptive cell therapy.

## 8. Concluding Remarks

Because of expanding knowledge on different T helper subsets and their functions in cancer biology, it is becoming increasingly clear that CD4^+^ T cells play a vital role in the complex interplay between the immune system and cancer. In this review, we summarized the current knowledge on the role of the different T helper subsets in tumorigenesis and their applications in adoptive cell therapy. The influence of each of these subsets in tumor progression is, in most cases, dependent on the cancer type and the dynamic balance with other players of the tumor microenvironment. With more evidence becoming progressively available, it will be interesting to understand whether differences in relevance exist between hematological and solid malignancies or if such variations are rather due to disease and patient heterogeneity. Therefore, a deeper understanding of how the different T helper cell subsets interact with tumors is crucial to uncover new therapeutic opportunities to use different T helper subsets for cancer treatment.

## Figures and Tables

**Figure 1 cancers-15-01650-f001:**
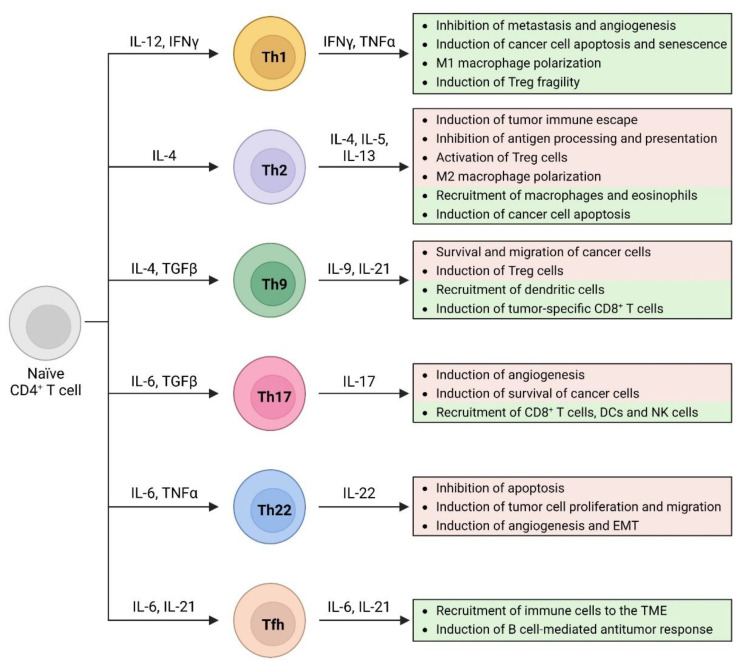
CD4^+^ T helper subsets and their functions in the antitumoral immune response. Depending on the cytokine milieu, naïve CD4^+^ T cells differentiate into different T helper subsets with different cytokine expression profiles and diverse functions in tumor immunology. Treg, T regulatory cells; DCs, dendritic cells; NK, natural killer; EMT, epithelial-to-mesenchymal transition; TME, tumor microenvironment.

**Figure 2 cancers-15-01650-f002:**
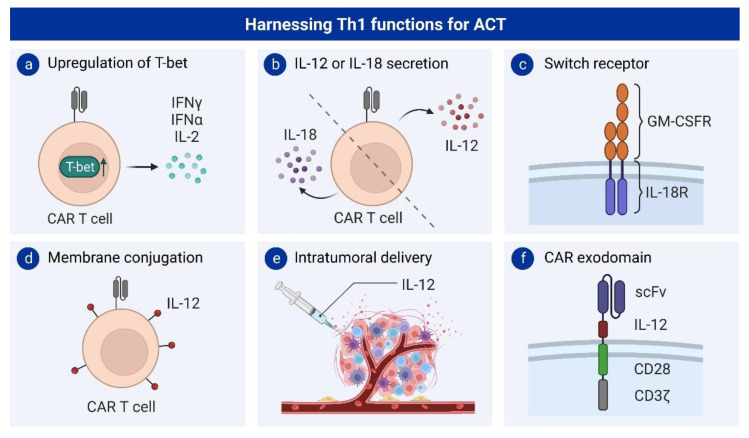
Strategies harnessing Th1 functions in ACT: (**a**) Upregulation of the Th1 transcription factor T-bet. (**b**) CAR T cells secreting IL-18 or IL12. (**c**) Chimeric GM-CSFR/IL-18R receptor. (**d**) Conjugation of IL-12 on the membrane of CAR T cells. (**e**) Intratumoral delivery of IL-12. (**f**) Integration of IL-12 in the CAR exodomain. scFv: single-chain variable fragment.

**Figure 3 cancers-15-01650-f003:**
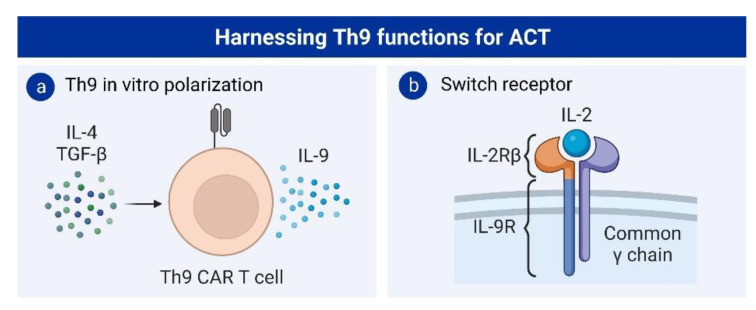
Strategies harnessing Th9 functions in ACT: (**a**) in vitro polarization of CAR T cells to IL-9-secreting T cells; and (**b**) chimeric receptor with the extracellular domain of the IL-2 receptor β (IL-2Rβ) and the IL-9R, which dimerizes with the common γ chain.

**Figure 4 cancers-15-01650-f004:**
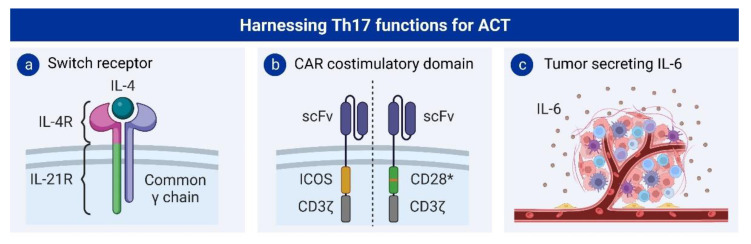
Strategies harnessing Th17 functions in ACT: (**a**) Chimeric cytokine receptor formed by the extracellular domain of the IL-4R and the intracellular domain of the IL-21R, which dimerizes with the common γ chain. (**b**) Using ICOS or mutating the CD28 costimulatory domain (CD28*) polarizes CAR T cells towards a Th17 phenotype. (**c**) Engineering of TGF-β-producing cancer cells to secrete IL-6 skews T-cell polarization towards Th17. scFv: short-chain variable fragment.

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
