# Peer review of "Role and Potential of Different T Helper Cell Subsets in Adoptive Cell Therapy"

_cancers, 2023, doi:10.3390/cancers15061650_

Round 1
Reviewer 1 Report
The authors have summarized the role of different T helper subsets in ACT therapy, which including Th1, Th2, Th9, Th17, Th22 and Tfh. In the all, this review is comprehensive and provides a thorough overview of the current state of research in the field. And some aspects should be described before acceptance.
1. In the tile, the author stated the promise of different T helper subsets in ACT. However, the promise of different T helper cell is relatively few.
2. In the parts of simple summary and abstract, there are too much description of background. In contrast, the introduction about the theme of this review is limited.
3. In the part of introduction, the description of Th cells in tumor immunity should be added.
4. The limitation of Th cells in ACT therapy should be added.
5. More clinical studies about the Th cells in ACT should be described.
Reviewer 2 Report
The manuscript from David Andreu-Sanz and Sebastian Kobold is a very well written, comprehensive and informative review. The sections and discussion are logically and clearly presented; the figures provided are clear and helpfully illustrate the concepts described in the text. The manuscript will provide a valuable reference for the field.
I have just one comment; the title and abstract suggest the review is focused on the role of CD4 T cells. However, the majority of studies described address the role of CD4 T cell derived cytokines and not direct effects of CD4 T cells themselves. It would perhaps be helpful to include a short paragraph to discuss whether the anti-tumour role of CD4 T cells goes beyond cytokine production (e.g. direct tumour killing) and any implication this has for the design of effective cell therapy approaches.
Reviewer 3 Report
This review summarizes the role of CD4+ T cells in cancer immunotherapy. Article is nicely written and summarized but does not add any new information. Can authors describe what new information they have provided in this review article compared to already published article (cited as reference 6 in this article). To add more information for the reader, my suggestion is to include one section discussing the limitations and advantages of CD4+ T cell therapy comparing hematological and solid malignancies.
